# Isolable zero-valent Ditin(0) and Diplumbum(0) complexes

Jinghang Shen [1], Zhengting Zhang[1], Xiaokang Ke[1], Luming Peng [1],
Qianyi Zhao[2], Congqing Zhu [1]✉ & Qin Zhu [1]✉

Although complexes with monatomic zero-valent main group centers have been reported, diatomic zero-valent complexes are extremely rare and all previously reported examples were stabilized by either carbene or silylene ligands. Here, we present the isolation of diatomic E(0)-E(0) (E = Sn, Pb) species supported by two [N{CH$_2$CH$_2$NP$^i$Pr$_2$}$_3$Sn] fragments. The reaction of trilithium salt N{CH$_2$CH$_2$NLiP$^i$Pr$_2$}$_3$ with SnCl$_2$ yields complex [N{CH$_2$CH$_2$NP$^i$Pr$_2$}$_2$]$_2$Sn$_3$ (**1**) with a Sn$_3$ chain. The reduction of the mixture of **1** and SnCl$_2$ with KC$_8$ produces the catenated Sn$_4$ chain [N{CH$_2$CH$_2$NP$^i$Pr$_2$}$_3$Sn$_2$]$_2$ (**2**), featuring a diatomic Sn(0)-Sn(0) unit. Further reduction of **2** with KC$_8$ yields the alkali metal ion-bridged complex [N{CH$_2$CH$_2$NP$^i$Pr$_2$}$_3$SnK]$_2$ (**3**). Moreover, the reaction of **3** with PbI$_2$ and KC$_8$ affords [N{CH$_2$CH$_2$NP$^i$Pr$_2$}$_3$SnPb]$_2$ (**4**), which can also be generated by the reaction of KC$_8$ with PbI$_2$ and [N{CH$_2$CH$_2$NP$^i$Pr$_2$}$_3$SnLi]$_2$ (**5**). Complex **4** features a diatomic Pb(0)-Pb(0) unit, representing a heavy diatomic zero-valent main group complex. The presence of diatomic E(0)-E(0) (E = Sn, Pb) units in complexes **2** and **4**, respectively, is further confirmed by computational studies.

Low-valent main-group complexes have garnered increasing attention from both theoretical and experimental chemists due to their intriguing electronic structures and diverse reactivities[1–4]. Based on quantum chemical analysis, Frenking and co-workers discovered that the bonding in carbodiphosphorane can be described as a central carbon(0) atom complexed by two donor ligands[5,6]. Subsequently, a series of authentic monoatomic zero-valent group 14 complexes have been experimentally isolated[7–30]. However, the isolation of diatomic zero-valent group 14 complexes featuring element-element bonds is markedly challenging due to their inherent instability.

In 2008, Robinson and co-workers reported a remarkable diatomic silicon(0) complex (**I** in Fig. 1), [NHC→Si=Si←NHC], featuring a Si=Si double bond and supported by bulky N-heterocyclic carbenes (NHCs)[31]. This complex was synthesized through the reduction of [NHC→SiCl$_4$] with KC$_8$ and was described as a soluble "allotrope" of silicon[32]. Following this pioneering work, Jones and co-workers demonstrated that heavier group 14 analogs [NHC→E=E←NHC] (E = Ge, Sn) could be

synthesized by reducing the corresponding [NHC→ECl$_2$] with a magnesium(I) dimer instead of KC$_8$ (**I** in Fig. 1)[33,34]. In 2014, So and co-workers reported a digermanium(0) complex supported by N-heterocyclic silylene (NHSi) ligands (**II** in Fig. 1) from reducing [NHSi→GeCl$_2$] with KC$_8$[35]. More recently, Mo and co-workers reported the synthesis and characterization of disilicon(0) and ditin(0) unit complexes, which were supported by an innovative N-heterocyclic imino-substituted NHSi ligand (**III** in Fig. 1)[36,37]. Employing modified silylene-carborane ligands, Driess and colleagues reported a Ge$_4$ cluster featuring a digermanium(0) moiety (**IV** in Fig. 1), in which the Ge$_4$ chain was described as Ge(III)−Ge(0)−Ge(0)−Ge(III) bonds[38].

These pioneering studies suggest that σ-donating ligands are essential for stabilizing diatomic E(0) complexes. To date, all isolated diatomic zero-valent group 14 complexes have been stabilized by NHC or NHSi ligands (Fig. 1). Consequently, the development of different ligands is highly beneficial for exploring the chemistry of zero-valent group 14 elements, as these ligands could induce different structural

[1]State Key Laboratory of Coordination Chemistry, Jiangsu Key Laboratory of Advanced Organic Materials, School of Chemistry and Chemical Engineering, Nanjing University, Nanjing 210023, China. [2]School of Chemistry and Chemical Engineering, Henan Normal University, Xinxiang, Henan 453007, China. ✉e-mail: zcq@nju.edu.cn; zhuqin@nju.edu.cn

**Fig. 1 | Examples of group 14 diatomic E(0) complexes.** Reported examples of group 14 diatomic complexes **I** – **IV** are stabilized by NHC or NHSi ligands (NHC: N-heterocyclic carbene; NHSi: N-heterocyclic silylene; Dipp = 2,6-$^i$Pr$_2$-C$_6$H$_3$). The Sn(0)-Sn(0) and Pb(0)-Pb(0) complexes **V** and **VI** supported by the double layer N-P ligand presented in this study.

and electronic properties. In recent years, we have developed a double layer nitrogen-phosphorus ligand, which plays crucial roles not only in constructing metal-metal bonds but also in activating small molecules[39–49]. Inspired by the isolation of complex **IV**, we believed that in the N{CH$_2$CH$_2$NHP$^i$Pr$_2$}$_3$ ligand, the N atoms (hard bases) can bind to the E(III) center, while the P atoms (soft bases) can bind to the E(0) center. Based on this double-layer N-P ligand, we hereby report the synthesis, characterization, and theoretical analysis of complexes featuring ditin(0) and diplumbum(0) units (**V** and **VI** in Fig. 1), which represent the examples of diatomic zero-valent group 14 complexes without NHC or NHSi ligands.

## Results

### Synthesis of Sn$_3$ and Sn$_4$ chain complexes

Treatment of trilithium salt[39] N{CH$_2$CH$_2$NLiP$^i$Pr$_2$}$_3$ with 1.5 equivalents of SnCl$_2$ at room temperature for overnight resulted in the isolation of complex [N{CH$_2$CH$_2$NP$^i$Pr$_2$}$_3$]$_2$Sn$_3$ (**1**) as yellow crystals with a yield of 76% (Fig. 2). Using more equivalents of SnCl$_2$ did not alter the product of this reaction. The structure of **1** was elucidated through single-crystal X-ray diffraction, nuclear magnetic resonance (NMR) spectroscopy, UV-Vis absorption spectroscopy, and elemental analysis (EA). In the $^{31}$P{$^1$H} NMR spectrum of complex **1**, a singlet resonance at $\delta_P = 67.5$ ppm indicates equivalence of the three P atoms in solution on the NMR scale. The $^1$H NMR spectrum displays four resonances between $\delta_H = 3.10$ and $1.15$ ppm, consistent with the three-fold symmetry of this species. This result suggests that the coordination between the P atoms and central Sn(0) atom is dynamically balanced in solution.

As shown in Fig. 3a, complex **1** features a bent Sn$_3$ chain. The central Sn(0) is coordinated with two P atoms (P1 and P1') and is also connected to Sn1 and Sn1'. The bond distance of Sn1−N1 (2.213(3) Å) is slightly longer than those of Sn1−N2 (2.143(3) Å) and Sn1−N3 (2.141(3) Å), but considerably shorter than that of Sn1−N4 (2.483(3) Å). The Sn−Sn2 bond length is 3.1115(6) Å, which is longer than the sum of the covalent radii for a single bond between two Sn atoms (2.80 Å)[50]. The formal shortness ratio (FSR), defined as the ratio of the M−M bond

length to the sum of the covalent atomic radii of the two metals, for the Sn−Sn bond is 1.11. This Sn−Sn bond length and FSR value are greater than those observed in structurally authenticated Sn−Sn single bonds in Sn$_3$ complexes [Sn-C$_6$H$_3$(CH$_2$N$^i$Pr$_2$)$_2$-2,6]$_3$H (2.930(2) Å, FSR = 1.05; 2.921(2) Å, FSR = 1.04)[51] and [2,5-(2'-C$_5$H$_4$N)$_2$-3,4-(C$_6$H$_5$)$_2$-C$_4$N-Sn]$_3$Cl (2.8521(6) Å, FSR = 1.02; 2.8632(6) Å, FSR = 1.02)[52], but shorter than the 2c-1e Sn•••Sn bond in [K(THF)$_6$][($o$-C$_6$H$_4$(2',6'-$^i$Pr$_2$C$_6$H$_3$N)$_2$)Sn•••Sn($o$-C$_6$H$_4$(2',6'-$^i$Pr$_2$C$_6$H$_3$)$_2$N)] (3.2155(9) Å, FSR = 1.15)[53]. The bond angle of Sn1−Sn2−Sn1' is 149.55(2)°, which is in the range of angles reported for Sn$_3$ complexes (145.070(16)° to 156.01(3)°)[24,52]. Formally, the oxidation state of the Sn atoms in **1** is Sn(III)−Sn(0)−Sn(III), indicating that Sn(II) from SnCl$_2$ undergoes a self-redox reaction in this process.

As one of the SnCl$_2$ molecules was reduced to Sn(0) in the formation of complex **1**, we investigated the reaction of N{CH$_2$CH$_2$NLiP$^i$Pr$_2$}$_3$ with SnCl$_2$ and KC$_8$ in THF. This reaction yielded a new catenated Sn$_4$ chain complex [N{CH$_2$CH$_2$NP$^i$Pr$_2$}$_3$Sn$_2$]$_2$ (**2**), which was isolated as a red crystalline solid with a 30% yield (Fig. 2). Complex **2** can be formed via a one-pot reaction of complex **1** with 1 equivalent of SnCl$_2$ and 2 equivalents of KC$_8$ in THF. In contrast with complex **1**, the $^1$H NMR spectrum of complex **2** displays resonances in a 4:8:8:4:4:8:72 ratio, and the $^{31}$P{$^1$H} NMR spectrum of complex **2** shows two sets of signals at $\delta_P = 67.5$ and $64.3$ ppm in a 2:1 ratio, indicating that the coordination environments among the three arms are distinct in solution. Despite numerous attempts, obtaining liquid-state $^{119}$Sn NMR spectra for complexes **1** and **2** was unsuccessful. However, solid-state NMR experiments yielded signals with large $^{119}$Sn chemical shift anisotropy. For example, in complex **2**, the single-pulse NMR spectrum reveals three sets of signals at -42, -53, and -296 ppm (Supplementary Fig. S8). The combined intensity of the first two peaks is comparable to that of the resonance at -296 ppm. In the $^1$H→$^{119}$Sn CP-MAS NMR spectrum (Supplementary Fig. S8), the intensity of the peak at -296 ppm is significantly enhanced compared to the resonances at -42 and -53 ppm, indicating that this peak is associated with Sn species that have stronger Sn-H dipolar coupling. Therefore, the peak at -296 ppm can be tentatively assigned to the Sn1 and Sn4 atoms, while the peaks

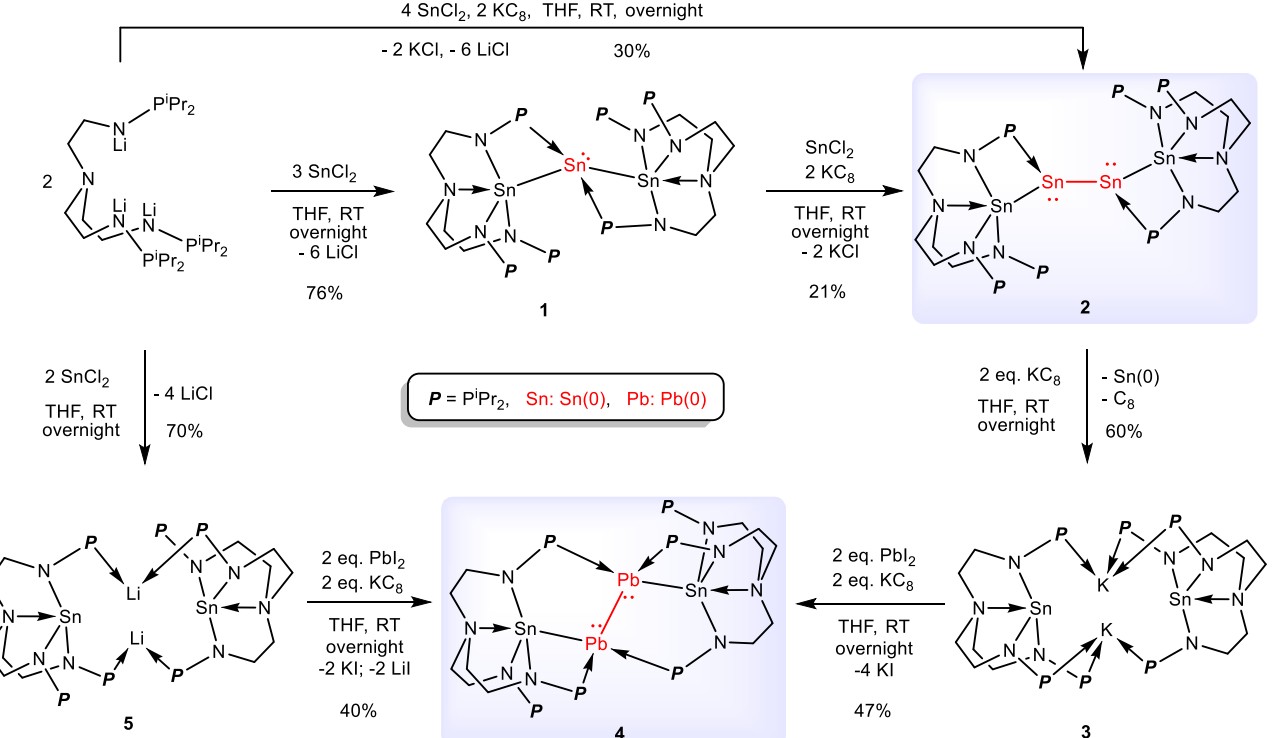

**Fig. 2 | Synthesis of diatomic Sn(0) and Pb(0) complexes.** Treatment of the trilithium salt N{CH$_2$CH$_2$NLiP$^i$Pr$_2$}$_3$ with SnCl$_2$ leads to the formation of the Sn$_3$ chain complex **1**, which can react with SnCl$_2$ and KC$_8$ to yield complex **2**, featuring a Sn(0)-Sn(0) bond. Complex **2** can be further reduced by KC$_8$ to form complex **3**. Complex **4**, featuring a Pb(0)-Pb(0) bond, is formed by the reaction of complex **3** with PbI$_2$ in the presence of KC$_8$. On the other hand, complex **4** could also be generated by the reaction of KC$_8$ with PbI$_2$ and [N{CH$_2$CH$_2$NP$^i$Pr$_2$}$_3$SnLi]$_2$ (**5**), which was formed by the reaction of trilithium salt with SnCl$_2$. RT = room temperature, THF = tetrahydrofuran.

at -42 and -53 ppm can be attributed to the Sn2 and Sn3 sites. This is because Sn1 and Sn4 have 8 hydrogen atoms in their third coordination shell, whereas Sn2 and Sn3 have only one hydrogen atom each. The peak splitting of the latter two peaks (approximately 1600 Hz) is likely due to complex J-coupling between the central Sn species, which is consistent with previous literature[54]. Additionally, this assignment of the peak at -296 ppm to Sn1 and Sn4 species in complex **2** aligns with the observation of strong signals at similar frequencies (centered at -228 ppm) in the single-pulse $^{119}$Sn NMR spectrum of complex **1** (Supplementary Fig. S4). However, attempts to acquire $^{119}$Sn solid-state NMR spectra of complexes **3**, **4** and **5** were unsuccessful, possibly due to the absence of direct Sn-Sn bonding between the two Sn sites, which may result in a long longitudinal relaxation time ($T_1$), a characteristic commonly observed in many Sn-containing compounds[55].

The UV-Vis absorption spectrum of complex **2** in THF shows a broad absorption at 489.0 nm, consistent with the complex's red color in solution. According to the results of time-dependent density functional theory (TD-DFT) calculations, the observed peak of the absorption wavelength ($\lambda_{exp}$ = 489.0 nm, $\lambda_{TD-DFT}$ = 462.8 nm, Supplementary Table S5) is primarily attributed to the π-π* excitation from the highest occupied molecular orbital (HOMO) to the lowest unoccupied molecular orbital (LUMO) in the S1 state, with a contribution of 92.6%.

The structure of complex **2** was also confirmed by single-crystal X-ray diffraction analysis (Fig. 3b), revealing a catenated Sn$_4$ chain between two [N{CH$_2$CH$_2$NP$^i$Pr$_2$}$_3$] ligands. The Sn2−Sn3 bond links two four-membered NPSn$_2$ rings to form an Sn$_4$ chain with Sn(III)−Sn(0)−Sn(0)−Sn(III) bonds; each of the Sn(III) atoms is bonded to three N atoms from the ligand. The central Sn2 and Sn3 atoms exhibit a pyramidal geometry, suggesting the presence of a lone pair of electrons at each Sn(0) atom. The Sn2−Sn3 distance of 2.9285(4) Å is slightly longer than the Sn(III)−Sn(0) distances of 2.9030(4) and 2.9173(4) Å. The

Sn−Sn bond lengths in complex **2** are shorter than those in complex **1** (3.1115(6) Å) and fall within the normal range reported for Sn−Sn single bonds. The FSR for the Sn−Sn bonds in complex **2** range from 1.04 to 1.05, consistent with the Sn−Sn bond in the reported complex [Sn-C$_6$H$_3$(CH$_2$N$^i$Pr$_2$)$_2$-2,6)$_4$H$_2$] (2.866(1) Å, FSR = 1.02; 2.776(1) Å, FSR = 0.99)[51]. The bond distances of P3−Sn2 (2.6698(13) Å) and P6−Sn3 (2.6772(13) Å) are similar to those in complex **1** (2.6594(9) Å). The bond angles of Sn1−Sn2−Sn3 and Sn2−Sn3−Sn4 are 91.730(19)° and 92.623(19)°, respectively, which are significantly smaller than the Sn1−Sn2−Sn1′ angle (149.55(2)°) in complex **1**. The dihedral angle between Sn1−Sn2−Sn3 and Sn2−Sn3−Sn4 is 1.50(3)°, indicating that the four Sn atoms are nearly coplanar.

## Synthesis of Pb(0)-Pb(0) complex

Complex **2** can be further reduced with alkali metals. Treating complex **2** with two equivalents of KC$_8$ at room temperature overnight yields the potassium-bridging complex **3** with 60% yield, along with elemental tin (Fig. 2). Complex **3** was fully characterized by NMR spectroscopy, elemental analysis, and UV-vis absorption spectra, and we further confirmed its structure through single-crystal X-ray diffraction (Fig. 3c). The $^1$H NMR spectrum of complex **3** shows four resonances at $\delta_H$ = 3.10, 2.43, 1.86, and 1.01 ppm in a ratio of 6:6:6:36, which is in line with a three-fold symmetric structure in solution. The $^{31}$P{$^1$H} NMR spectrum exhibits a single resonance with two sets of satellite peaks at $\delta_P$ = 54.0 ppm, suggesting the same coordination environment for all three phosphorus atoms.

Interestingly, treating complex **3** with two equivalents of PbI$_2$ and KC$_8$ in THF at room temperature produces a dark-purple solution. The binuclear Pb(0) complex **4** was isolated as dark purple crystals in 47% yield after recrystallization (Fig. 2). On the other hand, treatment of trilithium salt N{CH$_2$CH$_2$NLiP$^i$Pr$_2$}$_3$ (ref. 39) with one equivalent of SnCl$_2$ at room temperature for overnight resulted in the isolation of

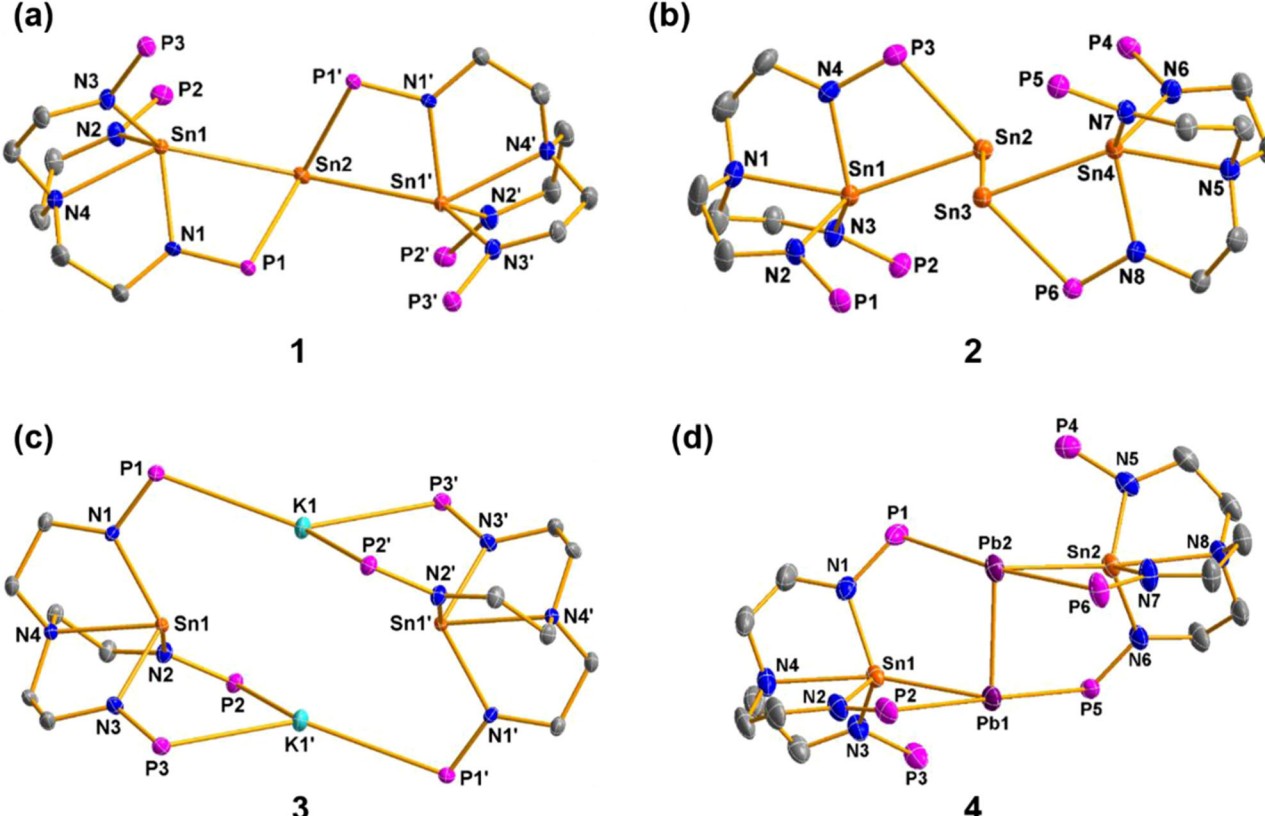

**Fig. 3 | Molecular structures of 1–4.** Solid-state structures of **1** (**a**), **2** (**b**), **3** (**c**) and **4** (**d**) by X-ray crystallography with 50% probability ellipsoids. Solvent molecules, hydrogen atoms and the isopropyl moieties in $P^iPr_2$ are omitted for clarity. Tin, orange; Plumbum, violet-red; Potassium, cyan; Phosphorus, pink; Nitrogen, blue; and Carbon, gray.

complex $[N\{CH_2CH_2NP^iPr_2\}_3SnLi]_2$ (**5**) as colorless crystals with a yield of 70% (Fig. 2). Complex **5** could react with two equivalents of $PbI_2$ and $KC_8$ in THF at room temperature, leading to the formation of complex **4** as dark purple crystals in 40% yield after recrystallization. Attempts to synthesize the complex with a $Pb_4$ unit by reacting $N\{CH_2CH_2NLiP^iPr_2\}_3$ with $PbI_2$ and subsequent treatment with $PbI_2$ and $KC_8$ were unsuccessful. Complex **4** was characterized by NMR spectroscopy, single-crystal X-ray diffraction analysis, UV-vis absorption spectra, and elemental analysis. The $^1H$ NMR spectrum of complex **4** presents four resonances at $\delta_H = 3.18$, 2.56, 2.19, and 1.15 ppm in a 12:12:12:72 ratio, and the $^{31}P\{^1H\}$ NMR of **4** reveals a single resonance with two sets of satellite peaks at $\delta_P = 67.3$ ppm, indicating a dynamic equilibrium in the coordination between phosphorus and lead atoms in solution. Attempts to collect the $^{119}Sn$ NMR spectra for complexes **3**, **4**, and **5** were unsuccessful both in solution and in the solid state, despite trying various conditions.

As exhibited in Fig. 3d, complex **4** features a bent $Sn_2Pb_2$ unit and the Pb(0)–Pb(0) unit was stabilized by two $[N\{CH_2CH_2NP^iPr_2\}_3Sn]$ moieties. The lead atoms exhibited a distorted tetrahedral geometry, as evidenced by the single-crystal X-ray diffraction analysis. One of the lead atoms is disordered over two sites in 97.8% and 2.2% occupancies. The major component having a Pb–Pb distance of 3.0452(6) Å and the Sn–Pb–Pb angle of 86.95(2)° and 86.99(2)°. The Pb-Pb distance is significantly longer than the sum of the two lead atoms' covalent single bond radii (2.88 Å) and the typical Pb–Pb single bond in diplumbanes, such as 2.844(4) Å in $Ph_3PbPbPh_3$[56]. However, the Pb-Pb distance of **4** is close to the Pb–Pb distance in reported diplumbene (such as 3.0515(3) Å in $Trip_2Pb=PbTrip_2$ (Trip = $2,4,6$-$Pr^i_3C_6H_2$))[57]. The Sn-Pb distances are 2.9589(7) Å for Sn1-Pb1 and 2.9367(6) Å for Sn2-Pb2. These structural parameters of complex **4** are reminiscent of the terphenyl substituted diplumbynes reported by Power and co-workers. In their examples, the Pb−Pb bond lengths ranging from 3.0382(6) to 3.1881(1) and the C−Pb−Pb angles ranging from 94.26(4) to 116.02(6)°[58,59]. The larger trans-bent angle results in multiple-bonding character with bond orders up to 1.5. The smaller angle (less than 90°) suggests that compound **4** contains a Pb-Pb single bond and two nonbonded pairs in each of lead centers. The Pb-Pb single bond is a result of head-to-head overlap of a 6p orbital from each lead atom which generates an unusually long Pb−Pb bond (3.0452(6) Å). This result is consistent with decreasing hybridization of the s and p orbitals in heavier main group elements[60]. Although previously reported E(0) complexes exhibit abundant reactivity[27,61], no reactions were observed between complexes **2** and **4** and a range of reagents, such as $Fe_2(CO)_9$, $Mn_2(CO)_{10}$, $Ph_2CO$, $^tBuN=C=O$, $^tBuNC$, $CuI$, $BPh_3$, CO, $CO_2$, and $N_2O$. This is likely due to the crowded environment around the Pb(0) or Sn(0) centers in these complexes. DFT calculations were carried out to investigate the coordination reactions between complexes **2** and **4** with molecules such as CO, THF, and $BCl_3$. The results showed that these reactions have relatively high reaction energies ($\Delta G > 0$ kcal/mol), indicating that they are endothermic processes and thus unlikely to occur (Supplementary Fig. S27).

## Theoretical studies

To gain insight into the nature of the bonding in diatomic Sn(0) and Pb(0) units in complexes **2** and **4**, DFT calculations were performed at the PBE0-D3BJ/6-311 + + G(d,p)-SDD//PBE0-D3BJ/6-31 g(d)-LanL2DZ level[62]. The DFT-calculated structures were found to be in good agreement with X-ray structural analyses (Supplementary Tables S3-S4). As shown in Fig. 4, HOMO for **2** and **4** were represented by σ bond

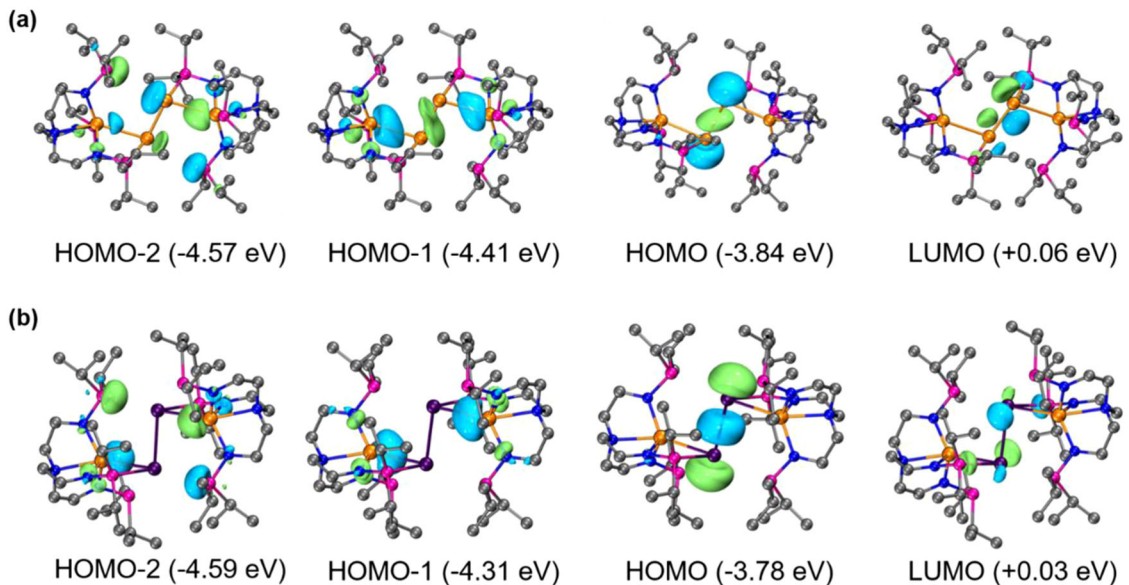

**Fig. 4 | Frontier molecular orbitals analysis.** Selected frontier molecular orbitals of **2** (**a**) and **4** (**b**) determined by DFT calculations at PBE0-D3BJ/6-311 + +g(d,p)-SDD// PBE0-D3BJ/6-31 g(d)-LanL2DZ level (isovalue = 0.06 a.u.). The HOMO orbitals of complexes **2** and **4** represent the lone pairs at the Sn(0) and Pb(0) centers.

for the E(0)−E(0) (E = Sn, Pb) atoms with major contributions *p* orbital from 30.5% for Sn(0) and 33.6% for Pb(0), whereas the LUMO represents the π* anti-boding of E(0)−E(0). In addition, the HOMO-1 and HOMO-2 described the Sn(III)−E(0) and P(III)−E(0) interactions (Fig. 4). The HOMO-LUMO gaps for complexes **2** and **4** were 3.90 and 3.81 eV, respectively. The natural population analysis (NPA)[63] of complex **2** on the Sn(III) and Sn(0) are 1.76 and -0.10, whereas for Sn(III) and Pb(0) in complex **4** are 1.71 and -0.06. This observation is consistent with the E(0) centers being electron-rich according to electrostatic potential and atomic dipole corrected Hirshfeld atomic charge analyse (Supplementary Fig. S28)[64].

The Wiberg bond index (WBI) of the Sn(0)−Sn(0) bond and the Pb(0)−Pb(0) bond are 0.91 and 0.90, respectively, consistent with single bond character. These results lead us to describe the structure of **2** and **4** most accurately as Sn(III)−E(0)−E(0)−Sn(III) (E = Sn or Pb) complexes. More interestingly, the lone pair electrons of Sn(0) and Pb(0) in complexes **2** and **4** interact with an adjacent hydrogen atom of P$^i$Pr$_2$ fragment, forming hydrogen bonds with bond lengths of 2.91 and 2.80 Å, respectively. The hydrogen bonding angles observed for P-Sn...H and P-Pb...H are 142.0⁰ and 165.2⁰ (Supplementary Fig. S29), indicating the strong directional nature of the hydrogen bonding interactions between the metal centers and the hydrogen atom in P$^i$Pr$_2$. Atoms in molecules (AIM)[64,65] analysis confirms the existence of Sn- and Pb-involving hydrogen bonds, characterized by bond critical points (BCPs) and bond paths connecting the Sn and Pb to the hydrogen atom. Additionally, the analysis reveals the presence of Pb−Pb, Pb−Sn, and Pb−P interactions (Fig. 5b, Supplementary Fig. S30). The C-H...Pb hydrogen bonds and Pb-P dispersion interactions were further characterized by noncovalent interactions (NCI) analysis (Fig. 5c, d)[66]. Such an interaction for C-H...Sn hydrogen bonds and Sn−P dispersion interactions were also observed in complex **2** (Supplementary Fig. S31).

The principal interacting orbital (PIO) analysis efficiently condenses complex delocalization interactions into a few semi-local orbitals, providing intuitive and interpretable chemical insights that are crucial for understanding the formation of chemical bonds[67,68]. The PIO analysis in this study examines the bonding modes of Sn(0) in complex **2** by dividing it into two fragments: the Sn(0) atom and the remaining structure (Fig. 6).

The Sn(0)-Sn(0) and Sn(III)-Sn(0) bonds correspond to σ-type interactions with PIO bond index (PBI) values of 1.00 and 0.97, respectively (Fig. 6a, b). The Sn(0) atom employs its 5p orbitals to form the dative P(III):→Sn(0) bonds with contributions of 1.59 *e* from P(III) and 0.41 *e* from Sn(0) (Fig. 6c). In addition, the Sn(0) has a non-bonding lone pair interacting with adjacent N and P vacancy orbitals with a PBI of 0.17. Such an interaction causes a decrease in the lone pair population from 2.00 to 1.91 *e* of Sn(0) (Fig. 6d). Furthermore, three interactions between Sn and adjacent atoms (two Sn and one P) were supported by natural adaptive orbital (NAdO) analysis[69] with close-to-one eigenvalues (0.81, 0.72, and 0.71, Supplementary Fig. S32).

Complex **4** was also analysis similarly, the Pb(0)−Pb(0) and Sn(III)−Pb(0) bonds correspond to σ-type interaction with PBI values of 1.00 and 0.95, respectively (Fig. 7). The PIO analysis of Pb(0) indicates that the Pb(0)−Pb(0) bond in **4** was formed by 6p valence orbitals of both Pb atoms with close-to-one value contributions of from two Pb(0) atoms (1.04 and 0.96 *e*, respectively, Fig. 7a). In the case of the Sn(III)−Pb(0) bonds in **4**, the Pb(0) atom involves a tangential 6p orbital to form a Sn(III)−Pb(0) bond with contributions of 1.23 *e* from Sn(III) and 0.77 *e* from Pb(0). The Pb(0) atom uses its 6p orbitals to form two Pb−P bonds with contributions of 0.40 *e* from Pb(0) and 1.60 *e* from P(III) (Fig. 7c), corresponding to a dative P(III):→Pb(0) bond. The Pb(0) also has a non-bonding lone pair interacting with adjacent N and P vacancy orbitals with a PBI of 0.11. Such an interaction causes a decrease in the lone pair population from 2.00 to 1.92 *e* of Pb(0) (Fig. 7d). Furthermore, these four primary interactions between Sn(III), Pb(0) and the two P(III) atoms were supported by NAdO analysis[69], where the key bonding of four σ-type NAdOs were located with the Eigenvalues of 0.78, 0.69, 0.68, and 0.16, respectively (Supplementary Fig. S33).

## Discussion

In summary, the complexes featuring diatomic zero-valent Sn(0)−Sn(0) and Pb(0)−Pb(0) units were stabilized by a double-layer N-P ligand. These complexes represent the examples of diatomic group 14 complexes without carbene or silylene ligands. Moreover, complex **4**, featuring a Pb(0)-Pb(0) unit, represents a heavy diatomic zero-valent main group complex. X-ray crystallography and theoretical studies indicate that these complexes are best considered

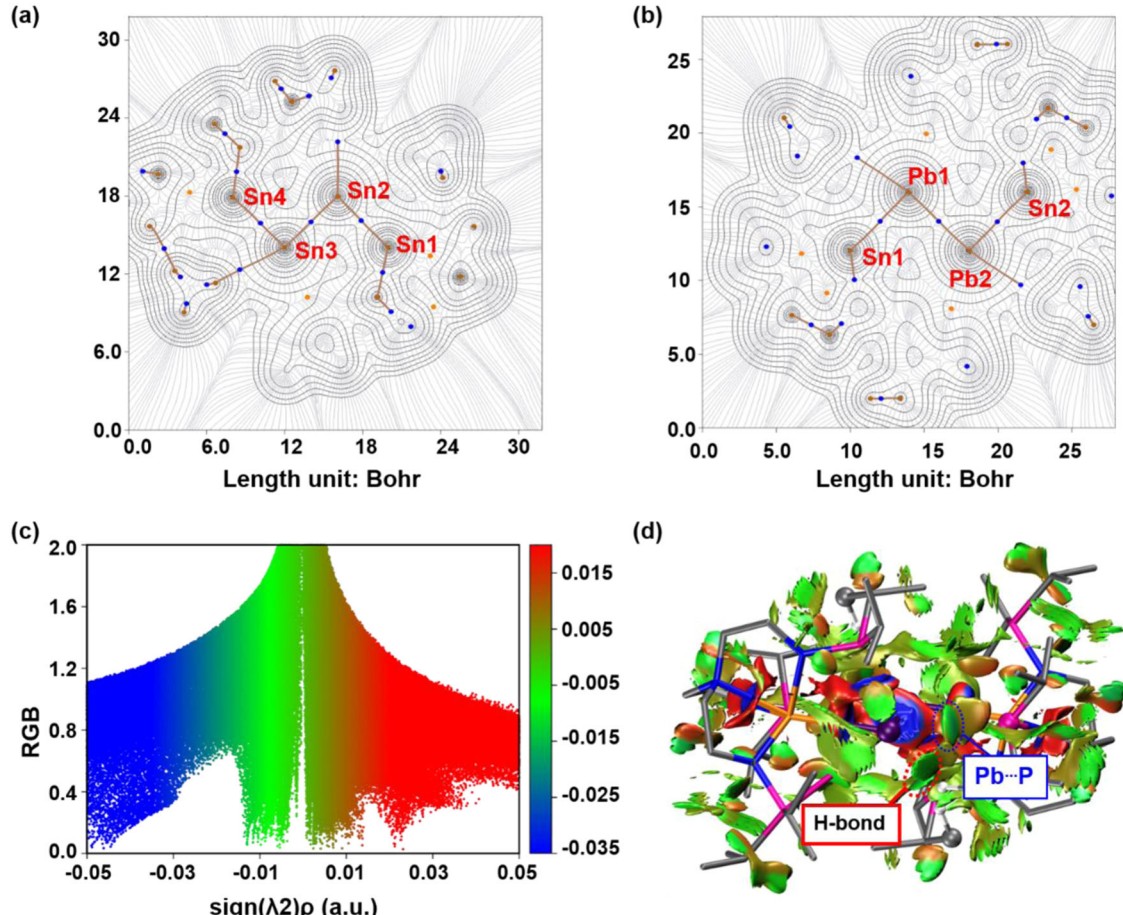

**Fig. 5 | Topological property analysis of complexes 2 and 4. a, b** Electron density gradient line map containing critical points and topological paths of complexes **2** and **4**, respectively. Brown, blue, and orange dots denote (3,-3), (3,-1) and (3,+1) critical points, respectively. **c** Visualization of the reduced density gradient (RDG) of complex **4** in relation to the sign($\lambda$2)$\rho$. **d** Noncovalent Interaction (NCI) analysis of complex **4**.

as Sn(III)−E(0)−E(0)−Sn(III) (E = Sn or Pb) and that lone pairs of electrons are observed at both internal E(0) atoms. The present findings suggest that carbene and silylene ligands are not indispensable for the stabilization of zero-valent group 14 entities. We are exploring the potential of this double layer N-P ligand for the preparation of other low-valent main group complexes.

## Methods

### General considerations

All manipulations were performed under an argon glovebox. Commercially available chemicals were used as received without further purification. Deuterated solvents were dried over Na/K (benzene-$d_6$, THF-$d_8$) and stored under an argon atmosphere prior to use. Nuclear magnetic resonance spectroscopy was performed using a Bruker AVIII-400 ($^1$H 400 MHz; $^{13}$C{$^1$H} 101 MHz; $^{31}$P{$^1$H} 162 MHz) at room temperature. The $^1$H and $^{13}$C{$^1$H} NMR chemical shifts ($\delta$) are relative to tetramethylsilane, and $^{31}$P{$^1$H} NMR chemical shifts are relative to 85% $H_3PO_4$. Absolute values of the coupling constants are provided in Hertz (Hz). Multiplicities are abbreviated as singlet (s), doublet (d), triplet (t), multiplet (m), and quartet (q). Solid-state $^{119}$Sn NMR spectra were recorded under conditions of magic angle spinning (MAS) at 9.4 T using a Bruker Avance III NMR spectrometer equipped with a 4.0 mm double tuned MAS probe. An excitation pulse of 1.7 µs, corresponding to a flip angle of $\pi$/2, and a recycle delay of 10 s were used in the $^{119}$Sn single pulse MAS NMR experiments. For $^1$H→$^{119}$Sn cross polarization (CP) MAS NMR experiment, a contact time of 3 ms and a recycle delay of 2 s were applied. $^1$H decoupling (rf power: 66 kHz) was used in the data acquisition for all experiments. Elemental analyses (C, H, N) were performed

on Vario MICRO cube elemental analyzer at the Center of Modern Analysis Nanjing University. UV-vis absorption spectra were collected at 25 °C with a UV3600. See the Supplementary Information for detailed spectra, crystallographic analyses, and computational details.

### Synthesis of N{CH₂CH₂NLiPⁱPr₂}₃[39]

Under stirring, a hexane solution of $^n$BuLi (0.6 mL, 2.4 M, 1.5 mmol, 3 equiv.) was added dropwise to a cold THF solution of N(CH₂CH₂NHPⁱPr₂)₃ (247 mg, 0.5 mmol, 1 equiv.). The mixture was stirred for 3 h and warmed up at room temperature. After removal of the volatiles under reduced pressure, the residue was extracted into hexane and filtered through a Celite-padded, coarse-porosity fritted filter. The filtrate was collected, and the volatiles were subsequently removed under reduced pressure. N{CH₂CH₂NLiPⁱPr₂}₃ was obtained as a white powder (251 mg, 98%). $^1$H NMR (400 MHz, THF-d₈, ppm) $\delta$ 3.23 (m, 6H, CH₂), 2.61 (m, 6H, CH₂), 2.07 (m, 6H, CH), 1.34 (m, 36H, CH₃).

### Synthesis of 1

Under stirring, a THF solution of SnCl₂ (142 mg, 0.75 mmol, 1.5 equiv.) was added to a THF solution of N(CH₂CH₂NLiPⁱPr₂)₃ (256 mg, 0.5 mmol, 1 equiv.). The mixture was stirred continuously overnight at room temperature. After removal of the volatiles under reduced pressure, the residue was extracted into hexane and filtered through a Celite-padded, coarse-porosity fritted filter. The filtrate was collected, and the volatiles were subsequently removed under reduced pressure. Complex **1** was obtained as a yellow powder (250 mg, 76%). Single crystals of **1** suitable for X-ray diffraction were grown from a THF solution at −30 °C. $^1$H NMR (THF-d₈, 400 MHz, ppm) $\delta$ 3.10 (m, 6H, CH₂), 2.54 (m, 6H, CH₂), 2.19 (br,

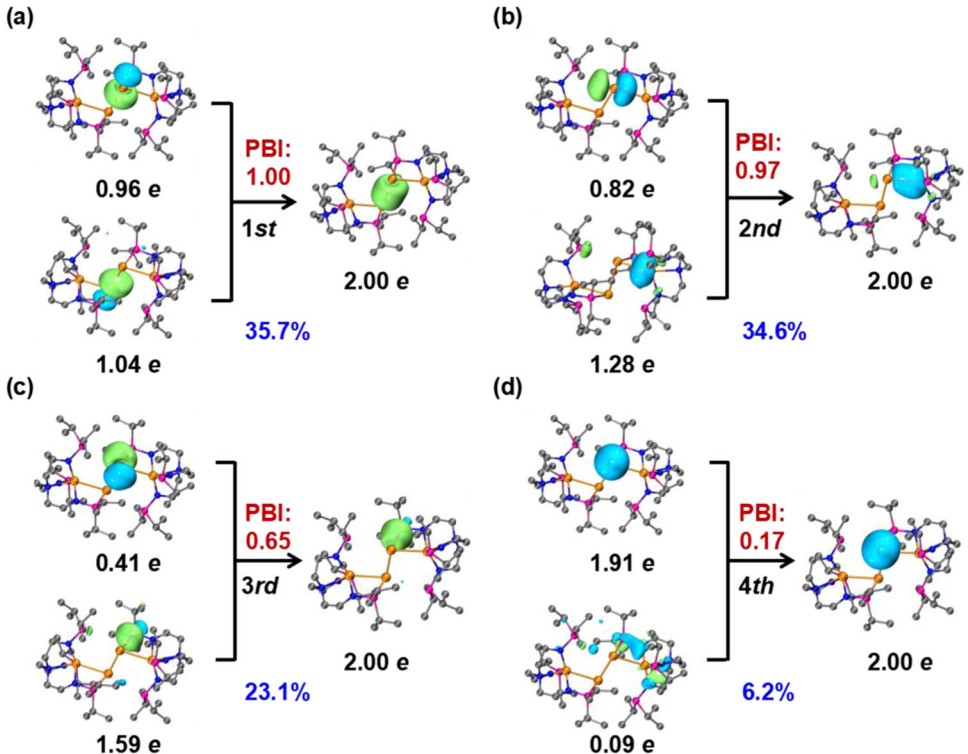

**Fig. 6 | Principal interacting orbital (PIO) analysis for complex 2.** Hydrogen atoms were omitted for clarity. Each pair (**a** - **d**) of PIOs results in the formation of a bonding PIMO (principal interacting molecular orbital). The strength of the interaction was assessed by the PBI (principal bonding index). The total PBI value of two fragments was 2.80. The isosurface 0.050 a.u. was plotted.

6H, CH), 1.15 (m, 36H, CH$_3$). $^{13}$C{$^1$H} NMR (THF-d$_8$, 101 MHz, ppm) δ 51.88 (s, NCH$_2$CH$_2$), 40.54 (d, $J$ = 9 Hz CH$_2$CH$_2$NP), 26.50 (d, $J$ = 9 Hz PCH), 20.01 (s, CH(CH$_3$)$_2$), 19.85 (s, CH(CH$_3$)$_2$). $^{31}$P{$^1$H} NMR (THF-d$_8$, 162 MHz, ppm) δ 67.5. Anal. calcd for C$_{48}$H$_{108}$N$_8$P$_6$Sn$_3$: C, 43.02; H, 8.07; N, 8.36; found C, 42.93; H, 8.07; N, 8.31.

## Synthesis of 2
*Method A*: While stirring, SnCl$_2$ (189 mg, 1.0 mmol, 2 equiv.) and KC$_8$ (68 mg, 0.5 mmol, 1 equiv.) were added to a THF solution of N(CH$_2$CH$_2$NLiP$^i$Pr$_2$)$_3$ (256 mg, 0.5 mmol, 1 equiv.). The suspension was stirred continuously overnight at room temperature and then filtered through a funnel lined with Kimwipes. The red filtrate was concentrated to approximately 3 mL and stored at -30 °C for two days. The product precipitated out of the solution, was washed with cold n-pentane, and dried under reduced pressure to yield complex **2** as dark red crystals (107 mg, 30%). *Method B*: While stirring, SnCl$_2$ (18.9 mg, 0.1 mmol, 1 equiv.) and KC$_8$ (27.0 mg, 0.2 mmol, 2 equiv.) were added to a THF solution of [N(CH$_2$CH$_2$NP$^i$Pr$_2$)$_3$]$_2$Sn$_3$ (**1**) (133.9 mg, 0.1 mmol, 1 equiv.). The suspension was stirred overnight at room temperature and then filtered through a funnel lined with Kimwipes. The red filtrate was concentrated to approximately 1 mL and stored at -30 °C for two days. The product precipitated from the solution, was washed with cold n-pentane, and dried under reduced pressure to yield complex **2** as dark red crystals (30 mg, 21%). $^1$H NMR (THF-d$_8$, 400 MHz, ppm) δ 3.13 (s, 4H, CH$_2$), 3.07 (m, 8H, CH$_2$), 2.56 (m, 8H, CH$_2$), 2.51 (s, 4H, CH$_2$), 2.17 (s, 4H, CH), 2.02 (s, 8H, CH), 1.18 (m, 72H, CH$_3$). $^{31}$P{$^1$H} NMR (THF-d$_8$, 162 MHz, ppm) δ 67.5, 64.3. Anal. calcd for C$_{48}$H$_{108}$N$_8$P$_6$Sn$_4$: C, 39.50; H, 7.47; N, 7.23; found C, 39.24; H, 7.51; N, 7.25.

## Synthesis of 3
While stirring, KC$_8$ (16.2 mg, 0.12 mmol, 2.4 equiv.) was added to a THF solution of [N(CH$_2$CH$_2$NP$^i$Pr$_2$)$_3$]$_2$Sn$_4$ (**2**) (72.7 mg, 0.05 mmol, 1 equiv.). The mixture was stirred overnight at room temperature. Afterwards, the mixture was filtered through a funnel lined with Kimwipes. The colorless

filtrate was concentrated to approximately 1 mL and stored at -30 °C for one day. The product precipitated from the solution, was washed with cold n-pentane, and dried under reduced pressure to yield compound **3** as colorless crystals (38 mg, 60%). $^1$H NMR (THF-d$_8$, 400 MHz, ppm) δ 3.10 (dt, $J$ = 6.0, 4.8 Hz, 6H, CH$_2$), 2.43 (t, $J$ = 5.6 Hz, 6H, CH$_2$), 1.86 (m, 6H, CH), 1.01 (m, 36H, CH$_3$). $^{13}$C{$^1$H} NMR (THF-d$_8$, 101 MHz, ppm) δ 54.93 (s, NCH$_2$CH$_2$), 42.36 (d, $J$ = 6 Hz CH$_2$CH$_2$NP), 26.78 (d, $J$ = 11 Hz PCH), 21.71 (d, $J$ = 5 Hz CH(CH$_3$)$_2$), 21.38 (d, $J$ = 12 Hz CH(CH$_3$)$_2$). $^{31}$P{$^1$H} NMR (THF-d$_8$, 162 MHz, ppm) δ 54.0 ($^2J_{117Sn-31P}$ = 906 Hz, $^2J_{119Sn-31P}$ = 946 Hz). Anal. calcd for C$_{48}$H$_{108}$N$_8$K$_2$P$_6$Sn$_2$: C, 44.39; N, 8.63; H, 8.38, found C, 44.25; N, 8.25; H, 8.43.

## Synthesis of 4
*Method A:* While stirring, PbI$_2$ (46.2 mg, 0.10 mmol, 2 equiv.) and KC$_8$ (13.5 mg, 0.10 mmol, 2 equiv.) were added to a THF solution of [N(CH$_2$CH$_2$NP$^i$Pr$_2$)$_3$]$_2$Sn$_2$K$_2$ (**3**) (64.8 mg, 0.05 mmol, 1 equiv.). The mixture was stirred overnight at room temperature. Afterwards, the mixture was filtered through a funnel lined with Kimwipes. The purple filtrate was concentrated to approximately 1 mL and stored at -30 °C for one day. The product precipitated from the solution, was washed with cold n-pentane, and dried under reduced pressure to yield compound **4** as dark purple crystals (38 mg, 47%). *Method B*: While stirring, PbI$_2$ (46.2 mg, 0.10 mmol, 2 equiv.) and KC$_8$ (13.5 mg, 0.10 mmol, 2 equiv.) were added to a THF solution of [N(CH$_2$CH$_2$NP$^i$Pr$_2$)$_3$]$_2$Sn$_2$Li$_2$ (**5**) (61.7 mg, 0.05 mmol, 1 equiv.). The mixture was stirred overnight at room temperature. Afterwards, the mixture was filtered through a funnel lined with Kimwipes. The purple filtrate was concentrated to approximately 1 mL and stored at -30 °C for one day. The product precipitated from the solution, was washed with cold n-pentane, and dried under reduced pressure to yield compound **4** as dark purple crystals (32 mg, 40%). $^1$H NMR (THF-d$_8$, 400 MHz, ppm) δ 3.62 (m, 4H, THF) 3.18 (t, $J$ = 5.0 Hz, 12H, CH$_2$), 2.56 (t, $J$ = 5.6 Hz, 12H, CH$_2$), 2.19 (m, 12H, CH), 1.77 (m, 4H, THF), 1.15 (m, 72H, CH$_3$). $^{13}$C{$^1$H} NMR (THF-d$_8$, 101 MHz, ppm) δ 52.94 (s, NCH$_2$CH$_2$), 43.35 (d, $J$ = 12 Hz CH$_2$CH$_2$NP), 28.38 (d, $J$ = 8 Hz PCH),

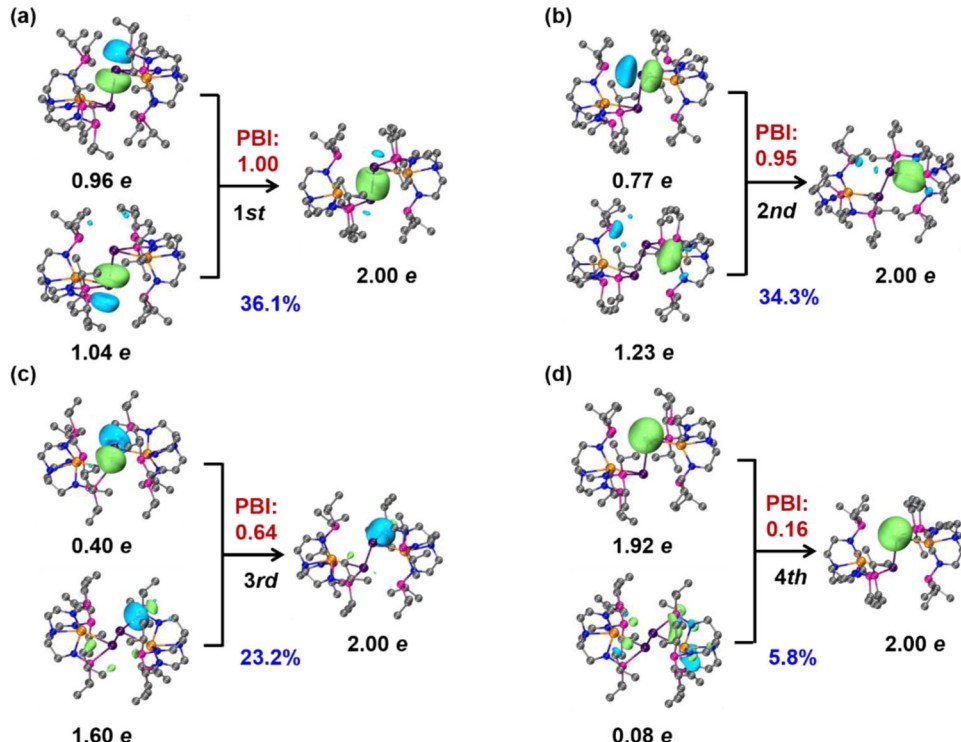

**Fig. 7 | Principal interacting orbital (PIO) analysis for complex 4.** Hydrogen atoms were omitted for clarity. Each pair (**a**–**d**) of PIOs results in the formation of a bonding PIMO. The strength of the interaction was assessed by the PBI. The total PBI value of two fragments was 2.76. The isosurface 0.050 a.u. was plotted.

21.60 (d, $J$ = 4 Hz CH(CH₃)₂), 21.35 (d, $J$ = 9 Hz CH(CH₃)₂). $^{31}P\{^1H\}$ NMR (THF-d₈, 162 MHz, ppm) δ 67.3 ($^1J_{Pb-P}$/$^2J_{Sn-P}$ = 365, 850 Hz). Anal. calcd for $C_{48}H_{108}N_8P_6Sn_2Pb_2(C_4H_8O)$: C, 36.48; N, 6.54; H, 6.78, found C, 36.50; N, 6.84; H, 7.08.

## Synthesis of 5

Under stirring, a THF solution of $SnCl_2$ (95 mg, 0.5 mmol, 1 equiv.) was added to a THF solution of $N(CH_2CH_2NLiP^iPr_2)_3$ (256 mg, 0.5 mmol, 1 equiv.). The mixture was stirred continuously overnight at room temperature. After removal of the volatiles under reduced pressure, the residue was extracted into hexane and filtered through a Celite-padded, coarse-porosity fritted filter. The filtrate was concentrated to approximately 3 mL and stored at -30 °C for two days. The product precipitated out of the solution, was washed with cold n-pentane, and dried under reduced pressure to yield complex **5** as colorless crystals (217 mg, 70%). $^1H$ NMR ($C_6D_6$, 400 MHz, ppm) δ 3.00 (m, 12H, $CH_2$), 2.24 (m, 12H, $CH_2$), 2.05 (m, 12H, CH), 1.25 (m, 72H, $CH_3$). $^{31}P\{^1H\}$ NMR (THF-d₈, 162 MHz, ppm) δ 51.4 (br), 49.6 (br). Anal. calcd for $C_{48}H_{108}N_8P_6Sn_2Li_2(C_4H_8O)_2$: C, 45.96; H, 8.07; N, 8.36; found C, 45.87; H, 8.28; N, 8.28.

## Data availability

All data are available from the corresponding author upon request. Crystal data of **1**, **2**, **3**, **4**, and **5** have been deposited at the Cambridge Crystallographic Data Center (CCDC) under reference numbers CCDC-2330637 (**1**), 2330638 (**2**), 2330636 (**3**), 2330639 (**4**), and 2377475 (**5**). These data can be obtained free of charge from The Cambridge Crystallographic Data Centre (www.ccdc.cam.ac.uk/data_request/cif). Source Data are included with this manuscript. Source data are provided with this paper.

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

## Acknowledgements

This research was supported by the National Key R&D Program of China (2021YFA1502500, C.Z.), the National Natural Science Foundation of China (Nos. 22401142, Q.Z. and 22271138, C.Z.), the Fundamental Research Funds for the Central Universities (020514380329, C.Z.), and the Open Research Fund of School of Chemistry and Chemical Engineering, Henan Normal University. High Performance Computing Center of Nanjing University is also acknowledged.

## Author contributions

Q.Z. conceived this project. J.S. performed the synthesis experiments. C.Z., Q.Z., and J.S. analyzed the experimental data. X.K., Z.Z., and L.P. performed the solid-state $^{119}$Sn NMR experiments and analyzed the data. Q.Z. conducted the theoretical computations. C.Z., J.S., and Q.Z. drafted the paper. All authors discussed the results and contributed to the preparation of the final manuscript.

## Competing interests

The authors declare no competing interests.
