## [Transparent Peer Review file · Nature Communications]

Isolable Zero-valent Ditin(0) and Diplumbum(0) Complexes

Corresponding Author: Professor Qin Zhu

Version 0:

Reviewer comments:

Reviewer #1

(Remarks to the Author)

This manuscript by Shen et al. presents the synthesis of Sn_3 , Sn_4 , and SnPb_2Sn units stabilized by a polydentate N/P ligand. The reported chemistry is intriguing, as such small chains of heavy metals with varying oxidation states are quite unusual. The computational study on the nature of the meta-metal bonds concludes that these complexes can be characterized as $\text{Sn(III)-E(0)-E(0)-Sn(III)}$ systems. This is a commendable piece of research that demonstrates that the stabilization of zero-valent group 14 moieties extends beyond carbenes and silylenes. The computational analysis is thorough, and the Natural Population Analysis (NPA) and Molecular Orbital (MO) results appear to substantiate the authors' claims regarding the metal centers. In this context, I find the quality and novelty of the manuscript suitable for publication in Nature Communications following minor revisions.

1. In the abstract, the authors state: "Here, we present the isolation of an unprecedented diatomic E(0)-E(0) ($\text{E} = \text{Sn, Pb}$) species supported by double layer nitrogen-phosphorus ligands." It appears that the E(0)_2 ($\text{E} = \text{Sn, Pb}$) units are stabilized by two $[\text{N}\{\text{CH}_2\text{CH}_2\text{N}(\text{Pr})_2\}_3\text{Sn}]$ fragments, rather than solely by the nitrogen-phosphorus ligands. Please clarify this point.
2. Please ensure that the font size in Figure 1 is uniform throughout for consistency.
3. Could the authors explore the synthesis of a complex with a Pb_4 unit by substituting the Sn atoms with Pb? I would appreciate if this reaction could be attempted.
4. Please specify the level of theory used in the DFT calculations within the main text.
5. Are the PIO results influenced by the choice of the fragments used? Please explain why the particular fragments presented in the article were selected.

Reviewer #2

(Remarks to the Author)

In this article titled Isolable Zero-valent Ditin(0) and Diplumbum(0) Complexes by Zhu and co-workers, the author has synthesized diatomic E(0)-E(0) ($\text{E} = \text{Sn, Pb}$) complexes supported by double-layer nitrogen-phosphorus ligands. This is interesting work, but it seems some additional work is required to understand the fundamental reactivity studies of isolated Sn(0) and Pb(0) complexes. Although the manuscript is well prepared, several serious concerns exist as mentioned below.

1. In the Introduction part, a discussion is needed on choosing a double-layer N-P ligand to stabilize the zero valent Sn (0) and Pb (0). The citation of key differences in terms of σ -donation and coordination with respect to the NHCs and NHSis is also recommended for a broad understanding.
2. Before NMR shift values, please mention the chemical shift (δ).
3. There are no ^{119}Sn NMR spectra for these compounds, and the authors mentioned that they could not record it due to poor solubility or could not detect it. However, solid-state NMR can be performed for all these complexes. Doing the solid-state NMR to understand the ^{119}Sn NMR shifts is highly recommended. The absence of ^{119}Sn NMR for all the complexes is not acceptable when the backbone is also having the tin atom.
4. The ^{119}Sn NMR of the reaction mixture can also be done. A comparison of the ^{119}Sn NMR values with the previous reports will be helpful to the readers.
5. The $^{13}\text{C}\{^1\text{H}\}$ NMR spectrum of 2 is also not measured.
6. In Figure 2, the yield for compound 4 is mentioned 47 %, whereas, lines 155 and 160 (page 9) described only 44% and 40%. The authors must recheck and correct it.
7. In Figure 2, there must be the liberation LiI while going from 5 to 4. The authors should correct the Scheme.

8. On page No7, mention the abbreviation of Tren iPPr ligand.
9. On Page no. 9, mention the geometry of the lead in the Pb(0)-Pb(0) complex.
10. Line 291, reference is missing for the synthesis of Trilithium salt $N\{CH_2CH_2NLiPiPr_2\}_3$.
11. It is highly recommended that some reactivity studies be performed with Pb(0) and Sn(0) complexes. Please check ref. a) *Angew. Chem. Int. Ed.* 61, e202114073 (2022) b) *Chem. Eur. J.* 2023, 29, e202203395.
12. The UV absorption spectra should be explained as only values are given without the mention of the electronic transitions, TDDFT can be helpful in this matter.

Reviewer #3

(Remarks to the Author)

This manuscript by Zhu and co-workers presents the synthesis and characterization of novel diatomic zero-valent E(0)-E(0) (E = Sn, Pb) complexes stabilized by a unique double-layer nitrogen-phosphorus ligand system. These compounds represent significant advances in zero-valent main group chemistry, especially complex 4, featuring a novel Pb(0)-Pb(0) unit. These species are well-characterized using a range of standard analytical methods, such as NMR, EA, UV-vis and X-ray, and their electronic structures are investigated by computational studies.

While the synthetic and structural aspects of the work are detailed and in-depth, the manuscript lacks crucial reactivity studies. Given that the computational analysis indicates the presence of lone pairs on the central E(0) atoms, it is essential for the authors to explore the reactivity of E(0)-E(0) (E = Sn, Pb) moieties. Investigations into nucleophilic behavior, as well as coordination with Lewis acids and interactions with small molecules (such as CO, CO₂ or H₂), could offer valuable insights.

Beyond these key concerns, the following specific points need to be addressed:

- 1) In lines 76 and 116, the chemical shifts for the $^{31}P\{^1H\}$ NMR spectra are reported with two decimal places, whereas in line 165, the ^{31}P NMR shift is reported with only one decimal place. Please standardize the format, preferably using one decimal place.
- 2) In line 194, there is a duplicate use of the word "the" that should be corrected.
- 3) In line 211, the description of the interaction with a hydrogen atom is unclear. Please specify the origin of this hydrogen.
- 4) The 1H NMR spectrum for compound 2, as presented in the Supplementary Information, is not sufficiently clean. A new spectrum should be recorded.
- 5) The ^{119}Sn NMR spectra for all new compounds are missing in the Supplementary Information. Could the authors provide an explanation for why the Sn signals were not observed? Have they attempted longer acquisition times or expanded the chemical shift range to obtain the signals?

In summary, I recommend the authors conduct additional reactivity studies to further probe the behavior of the Sn(0)-Sn(0) and Pb(0)-Pb(0) units, alongside addressing the specific points outlined above. Once these revisions are made, the manuscript will be in a better position for further consideration.

Version 1:

Reviewer comments:

Reviewer #1

(Remarks to the Author)

In this resubmitted manuscript, the authors did significant changes, to address the critical comments from both reviewers on the previous version. Therefore, acceptance is recommended.

Reviewer #2

(Remarks to the Author)

1. The fundamental reactivity studies of isolated Sn(0) and Pb(0) complexes is still elusive in this study. Although they have studied the energy profile of the complex 4 with a few small molecules including CO and BCl₃, but no reactivity study has been conducted with complex 2 either. This is very unusual that none of them reacting to anything, which is difficult to understand. As per the DFT calculation, some external energy source might work in this regard.
2. ^{119}Sn Solid-state NMR of the isolated complexes have been tried. Only complex 1 and 2 have been studied, but complexes 3, 4, and 5 are not observable. The authors need to write an explanation for the clarity.
3. The $^{13}C\{^1H\}$ NMR spectrum of 2 is also not measured. Authors can analyze solid-state ^{13}C NMR, if solution-state measurement is not possible.
4. The statement, "Attempts to synthesize the complex with a Pb₄ unit by the reaction of $N\{CH_2CH_2NLiPiPr_2\}_3$ with Pb₄

and subsequent treatment with Pbl₂ and KC₈ were unsuccessfully.” seems grammatically incorrect.

5. The reference for the level of theory for DFT calculations in the response sheet is 61, and in the manuscript it's 62.

Reviewer #3

(Remarks to the Author)

The authors have addressed most of the issues raised in the previous review and made substantial efforts to test the reactivity of Pb(0) and Sn(0) species, though no positive results were obtained. Their explanation, supported by computational analysis, reasonably attributes this to steric hindrance. With these modifications incorporated, I support the acceptance of the manuscript for publication.

Version 2:

Reviewer comments:

Reviewer #2

(Remarks to the Author)

The authors have addressed most of the comments raised by the reviewer and made substantial efforts to address them. With these modifications incorporated, I support the acceptance of the manuscript for publication.

Reviewer #1 (Remarks to the Author):

This manuscript by Shen et al. presents the synthesis of Sn_3 , Sn_4 , and SnPb_2Sn units stabilized by a polydentate N/P ligand. The reported chemistry is intriguing, as such small chains of heavy metals with varying oxidation states are quite unusual. The computational study on the nature of the meta-metal bonds concludes that these complexes can be characterized as $\text{Sn(III)-E(0)-E(0)-Sn(III)}$ systems. This is a commendable piece of research that demonstrates that the stabilization of zero-valent group 14 moieties extends beyond carbenes and silylenes. The computational analysis is thorough, and the Natural Population Analysis (NPA) and Molecular Orbital (MO) results appear to substantiate the authors' claims regarding the metal centers. In this context, I find the quality and novelty of the manuscript suitable for publication in Nature Communications following minor revisions.

Response: We sincerely thank the reviewer for the positive evaluation of our study. We have addressed all the issues raised in the revised manuscript and are grateful for the reviewer's suggestions, which have greatly contributed to the improvement of our paper.

1. In the abstract, the authors state: "Here, we present the isolation of an unprecedented diatomic E(0)-E(0) ($\text{E} = \text{Sn}, \text{Pb}$) species supported by double layer nitrogen-phosphorus ligands." It appears that the E(0)_2 ($\text{E} = \text{Sn}, \text{Pb}$) units are stabilized by two $[\text{N}\{\text{CH}_2\text{CH}_2\text{N}^i\text{Pr}_2\}_3\text{Sn}]$ fragments, rather than solely by the nitrogen-phosphorus ligands. Please clarify this point.

Response: We thank the reviewer for this excellent suggestion. To provide a more accurate description of the compound, we have revised the sentence in the abstract to read: "Here, we present the isolation of an unprecedented diatomic E(0)-E(0) ($\text{E} = \text{Sn}, \text{Pb}$) species supported by two $[\text{N}\{\text{CH}_2\text{CH}_2\text{N}^i\text{Pr}_2\}_3\text{Sn}]$ fragments."

2. Please ensure that the font size in Figure 1 is uniform throughout for consistency.

Response: Thanks for the remind. The font size in Figure 1 is uniformed in the revised version.

3. Could the authors explore the synthesis of a complex with a Pb_4 unit by substituting the Sn atoms with Pb? I would appreciate if this reaction could be attempted.

Response: We thank the reviewer for this excellent suggestion. According to the methods used for the synthesis of complexes with Sn_3 and Sn_4 chain, we attempted to synthesize the complexes with Pb_3 or Pb_4 units by reacting $\text{N}\{\text{CH}_2\text{CH}_2\text{N}^i\text{Pr}_2\}_3$ with PbI_2 and KC_8 under a series of conditions. However, despite our constant efforts, no desired species were isolated from these reactions.

Next, we tried to synthesize a complex with a Pb_4 unit using a method similar to the synthesis of complex Sn_2Pb_2 (complex **4**) from $[\text{N}\{\text{CH}_2\text{CH}_2\text{N}^i\text{Pr}_2\}_3\text{SnLi}]_2$ (complex **5**) with PbI_2 and KC_8 . We successfully isolated an analogous "ate" complex, $[\text{N}\{\text{CH}_2\text{CH}_2\text{N}^i\text{Pr}_2\}_3\text{PbLi}[\text{Li}(\text{THF})\text{I}]_2$, from the reaction of $\text{N}\{\text{CH}_2\text{CH}_2\text{N}^i\text{Pr}_2\}_3$ with PbI_2 . Unfortunately, treating this "ate" complex with PbI_2 and KC_8 resulted in a cloudy solution, with continuous formation of black powder even after filtering out the graphite and KC_8 . This is likely due to the extreme instability of the complex with the Pb_4 unit.

Although this result was negative, we have added the following sentence to the revised manuscript: "Attempts to synthesize the complex with a Pb_4 unit by the reaction of $\text{N}\{\text{CH}_2\text{CH}_2\text{N}^i\text{Pr}_2\}_3$ with PbI_2 and subsequent treatment with PbI_2 and KC_8 were unsuccessfully."

Figure R1. Synthesis and molecular structure of “ate” complex, $[N\{CH_2CH_2NP^iPr_2\}_3PbLi[Li(THF)]_2]$.

4. Please specify the level of theory used in the DFT calculations within the main text.

Response: Thank you for the suggestion. In the revised manuscript, we have added the computational details for geometry optimization. The following sentence has been included: “To gain insight into the nature of the bonding in diatomic Sn(0) and Pb(0) units in complexes **2** and **4**, DFT calculations were performed at the PBE0-D3BJ/6-311++G(d,p)~SDD//PBE0-D3BJ/6-31g(d)~LanL2DZ level.⁶¹” in the revised version.

5. Are the PIO results influenced by the choice of the fragments used? Please explain why the particular fragments presented in the article were selected.

Response: Thank you for the valuable suggestion. The PIO analysis focuses on identifying the primary interactions between two chemical fragments, which can be influenced by how the fragments are defined and chosen. In this work, we selected Sn(0) and Pb(0) as fragments to straightforwardly reflect the dominant bonding interactions for Sn(0) and Pb(0) in complexes **2** and **4**. The following sentence has been added to the revised manuscript: “The PIO analysis in this study examines the bonding modes of Sn(0) in complex **2** by dividing it into two fragments: the Sn(0) atom and the remaining structure (Fig. 6).”.

Reviewer #2 (Remarks to the Author):

In this article titled Isolable Zero-valent Ditin(0) and Diplumbum(0) Complexes by Zhu and co-workers, the author has synthesized diatomic E(0)-E(0) (E = Sn, Pb) complexes supported by double-layer nitrogen-phosphorus ligands. This is interesting work, but it seems some additional work is required to understand the fundamental reactivity studies of isolated Sn(0) and Pb(0) complexes. Although the manuscript is well prepared, several serious concerns exist as mentioned below.

Response: We sincerely thank the reviewer for the positive evaluation of our study. We have addressed all the issues raised in the revised manuscript and are grateful for the reviewer’s suggestions, which have greatly contributed to the improvement of our paper.

1. In the Introduction part, a discussion is needed on choosing a double-layer N-P ligand to stabilize the zero valent Sn (0) and Pb (0). The citation of key differences in terms of σ -donation and coordination with respect to the NHCs and NHSis is also recommended for a broad understanding.

Response: We thank the reviewer for this excellent suggestion. The following sentences have been added to the revised Introduction section: “Inspired by the isolation of complex **IV**, we believed that in the $N\{CH_2CH_2NHP^iPr_2\}_3$ ligand, the N atoms (hard bases) can bind to the E(III) center, while the P atoms (soft bases) can bind to the E(0) center.”.

2. Before NMR shift values, please mention the chemical shift (δ).

Response: The chemical shift label δ was added before NMR shift values in the revised manuscript.

3. There are no ^{119}Sn NMR spectra for these compounds, and the authors mentioned that they could not record it due to poor solubility or could not detect it. However, solid-state NMR can be performed for all these complexes. Doing the solid-state NMR to understand the ^{119}Sn NMR shifts is highly recommended. The absence of ^{119}Sn NMR for all the complexes is not acceptable when the backbone is also having the tin atom.

Response: We thank the reviewer for this excellent suggestion. All attempts to obtain liquid-state ^{119}Sn NMR spectra for these complexes were unsuccessful. Therefore, we examined the solid-state ^{119}Sn NMR spectra for these complexes as suggested. Fortunately, the solid-state ^{119}Sn NMR experiments for complexes **1** and **2** yielded signals with large ^{119}Sn chemical shift anisotropy, whereas no signals were detected for complexes **3**, **4**, and **5** despite trying various conditions in the solid state. The solid-state ^{119}Sn NMR spectra for complexes **1** and **2** have been added as Supplementary Figure S4 and Figure S8 in the revised Supplementary Information (SI), respectively. The ^{119}Sn NMR data for these complexes is also discussed in the revised manuscript as follows: "Despite numerous attempts, obtaining liquid-state ^{119}Sn NMR spectra for complexes **1** and **2** was unsuccessful. However, solid-state NMR experiments yielded signals with large ^{119}Sn chemical shift anisotropy. For example, in complex **2**, the single-pulse NMR spectrum reveals three sets of signals at -42, -53, and -296 ppm (Fig. S8c). The combined intensity of the first two peaks is comparable to that of the resonance at -296 ppm. In the $^1\text{H} \rightarrow ^{119}\text{Sn}$ CP-MAS NMR spectrum (Fig. S8a), the intensity of the peak at -296 ppm is significantly enhanced compared to the resonances at -42 and -53 ppm, indicating that this peak is associated with Sn species that have stronger Sn-H dipolar coupling. Therefore, the peak at -296 ppm can be tentatively assigned to the Sn1 and Sn4 atoms, while the peaks at -42 and -53 ppm can be attributed to the Sn2 and Sn3 sites. This is because Sn1 and Sn4 have 8 hydrogen atoms in their third coordination shell, whereas Sn2 and Sn3 have only one hydrogen atom each. The peak splitting of the latter two peaks (approximately 1600 Hz) is likely due to complex J-coupling between the central Sn species, which is consistent with previous literature.⁵⁴ Additionally, this assignment of the peak at -296 ppm to Sn1 and Sn4 species in complex **2** aligns with the observation of strong signals at similar frequencies (centered at -228 ppm) in the single-pulse ^{119}Sn NMR spectrum of complex **1** (Fig. S4)." in the revised version.

4. The ^{119}Sn NMR of the reaction mixture can also be done. A comparison of the ^{119}Sn NMR values with the previous reports will be helpful to the readers.

Response: We thank the reviewer for this suggestion. The ^{119}Sn NMR spectra of reaction mixture were examined, but no signals were obtained.

5. The $^{13}\text{C}\{^1\text{H}\}$ NMR spectrum of **2** is also not measured.

Response: We thank the reviewer for this suggestion. After recrystallization, complex **2** shows poor solubility, even in THF- d_8 , resulting in a low signal-to-noise ratio in its $^{13}\text{C}\{^1\text{H}\}$ NMR spectrum.

6. In Figure 2, the yield for compound **4** is mentioned 47%, whereas, lines 155 and 160 (page 9) described only 44% and 40%. The authors must recheck and correct it.

Response: Thanks for pointing this out, and we apologize for the mistake. The two methods yielded 47% and 40%, respectively. The yield has been corrected in the revised manuscript.

7. In Figure 2, there must be the liberation Lil while going from 5 to 4. The authors should correct the Scheme.

Response: Thanks for the reminder, and we apologize for this mistake. The scheme has been corrected in the revised manuscript.

8. On page No7, mention the abbreviation of Tren iPPr ligand.

Response: We thank the reviewer for this suggestion. The abbreviation for the Tren iPPr ligand has been added in the revised version. The revised sentence reads "..., revealing a catenated Sn₄ chain between two [N{CH₂CH₂NⁱPr₂}]₃ ligands."

9. On Page no. 9, mention the geometry of the lead in the Pb (0)-Pb (0) complex.

Response: We thank the reviewer for this excellent suggestion. The sentence "The lead atoms exhibited a distorted tetrahedral geometry, as evidenced by the single-crystal X-ray diffraction analysis." Has been added in the revised manuscript.

10. Line 291, reference is missing for the synthesis of Trilithium salt N{CH₂CH₂NLiⁱPr₂}₃.

Response: Thanks for the reminder. The reference for the synthesis of trilithium salt N{CH₂CH₂NLiⁱPr₂}₃ has been added to the revised manuscript.

11. It is highly recommended that some reactivity studies be performed with Pb(0) and Sn(0) complexes. Please check ref. a) Angew. Chem. Int. Ed. 61, e202114073 (2022) b) Chem. Eur. J. 2023, 29, e202203395.

Response: We thank the reviewer for this excellent suggestion. When we first isolated these Pb(0) and Sn(0) species, we attempted to study their reactivity. However, no reactions were observed between these complexes and a variety of reagents, such as metal carbonyls, ketones, metal halides, organoboron compounds, and other potential reactants. To gain further insight, we conducted DFT calculations and found that the reactions between complex 4 and small molecules are endothermic processes. Although this is not a positive result, we have added the following sentences to the revised manuscript: "Although previously reported E(0) complexes exhibit abundant reactivity,^{59,60} no reactions were observed between complexes 2 and 4 and a range of reagents, such as Fe₂(CO)₉, Mn₂(CO)₁₀, Ph₂CO, ^tBuN=C=O, ^tBuNC, CuI, BPh₃, CO, CO₂, and N₂O. This is likely due to the crowded environment around the Pb(0) or Sn(0) centers in these complexes. Performing density functional theory (DFT) calculations to investigate the coordination reactions between complex 4 and small molecules (CO, THF, and BCl₃) revealed that these reactions have relatively high reaction energies (ΔG > 0 kcal/mol), indicating endothermic processes that prevent spontaneous reaction (Fig. S26)". The detailed results have been included in the revised supplementary information as Supplementary Figure 26.

12. The UV absorption spectra should be explained as only values are given without the mention of the electronic transitions, TDDFT can be helpful in this matter.

Response: We appreciate the reviewer's comment. As suggested, the Time-Dependent Density

Functional Theory (TDDFT) calculations were employed to elucidate these electronic transitions, providing a deeper insight into the spectroscopic data. The following sentence has been added to the revised manuscript: “According to the results of time-dependent density functional theory (TD-DFT) calculations, the observed peak of the absorption wavelength ($\lambda_{\text{exp}} = 489.0$ nm, $\lambda_{\text{TD-DFT}} = 462.8$ nm, Table S10) is primarily attributed to the π - π^* excitation from the highest occupied molecular orbital (HOMO) to the lowest unoccupied molecular orbital (LUMO) in the S_1 state, with a contribution of 92.6%.”. The detailed results have been included in the revised supplementary information as Supplementary Table 10.

Reviewer #3 (Remarks to the Author):

This manuscript by Zhu and co-workers presents the synthesis and characterization of novel diatomic zero-valent E(0)-E(0) (E = Sn, Pb) complexes stabilized by a unique double-layer nitrogen-phosphorus ligand system. These compounds represent significant advances in zero-valent main group chemistry, especially complex 4, featuring a novel Pb(0)-Pb(0) unit. These species are well-characterized using a range of standard analytical methods, such as NMR, EA, UV-vis and X-ray, and their electronic structures are investigated by computational studies.

Response: We sincerely thank the reviewer for the positive evaluation of our study. We have addressed all the issues raised in the revised manuscript and are grateful for the reviewer’s suggestions, which have greatly contributed to the improvement of our paper.

While the synthetic and structural aspects of the work are detailed and in-depth, the manuscript lacks crucial reactivity studies. Given that the computational analysis indicates the presence of lone pairs on the central E(0) atoms, it is essential for the authors to explore the reactivity of E(0)-E(0) (E = Sn, Pb) moieties. Investigations into nucleophilic behavior, as well as coordination with Lewis acids and interactions with small molecules (such as CO, CO₂ or H₂), could offer valuable insights.

Response: We thank the reviewer for this excellent suggestion. When we first isolated these Pb(0) and Sn(0) species, we attempted to explore their reactivity. However, no reactions were observed between these Pb(0) or Sn(0) complexes and a range of reagents, such as metal carbonyls, ketones, metal halides, organoboron compounds, and other potential reactants. To gain further insight, we conducted DFT calculations and found that the reactions between complex 4 and small molecules are endothermic processes. Although this is not a positive result, we have added the following sentences to the revised manuscript: “Although previously reported E(0) complexes exhibit abundant reactivity,^{59,60} no reactions were observed between complexes 2 and 4 and a range of reagents, such as Fe₂(CO)₉, Mn₂(CO)₁₀, Ph₂CO, ^tBuN=C=O, ^tBuNC, CuI, BPh₃, CO, CO₂, and N₂O. This is likely due to the crowded environment around the Pb(0) or Sn(0) centers in these complexes. Performing density functional theory (DFT) calculations to investigate the coordination reactions between complex 4 and small molecules (CO, THF, and BCl₃) revealed that these reactions have relatively high reaction energies ($\Delta G > 0$ kcal/mol), indicating endothermic processes that prevent spontaneous reaction (Fig. S26)”. The detailed results have been included in the revised supplementary information as Supplementary Figure 26.

Beyond these key concerns, the following specific points need to be addressed:

1) In lines 76 and 116, the chemical shifts for the ³¹P{¹H} NMR spectra are reported with two

decimal places, whereas in line 165, the ^{31}P NMR shift is reported with only one decimal place. Please standardize the format, preferably using one decimal place.

Response: We thank the reviewer for this suggestion. The format of the chemical shifts for the $^{31}\text{P}\{^1\text{H}\}$ has been standardized to one decimal place in the revised version.

2) In line 194, there is a duplicate use of the word "the" that should be corrected.

Response: Thanks for the reminder, and we apologize for this mistake. The duplicate word "the" has been deleted in the revised manuscript.

3) In line 211, the description of the interaction with a hydrogen atom is unclear. Please specify the origin of this hydrogen.

Response: Thank you for the valuable suggestion. We have now included the geometric data related to the hydrogen bond and the corresponding angle concerning the hydrogen atom within the P^iPr_2 fragment that interacts with the $\text{Sn}(0)$ and $\text{Pb}(0)$ centers. The following sentences have been added to the revised manuscript: "More interestingly, the lone pair electrons of $\text{Sn}(0)$ and $\text{Pb}(0)$ in complexes **2** and **4** interact with an adjacent hydrogen atom of P^iPr_2 fragment, forming hydrogen bonds with bond lengths of 2.91 and 2.80 Å, respectively. The hydrogen bonding angles observed for $\text{P}-\text{Sn}\cdots\text{H}$ and $\text{P}-\text{Pb}\cdots\text{H}$ are 142.0° and 165.2° (Fig. S28), indicating the strong directional nature of the hydrogen bonding interactions between the metal centers and the hydrogen atom in P^iPr_2 ." The detailed results have been added to the revised supplementary information as Supplementary Figure 26.

4) The ^1H NMR spectrum for compound **2**, as presented in the Supplementary Information, is not sufficiently clean. A new spectrum should be recorded.

Response: We thank the reviewer for these suggestions. We have re-collected the ^1H NMR spectrum of complex **2**.

5) The ^{119}Sn NMR spectra for all new compounds are missing in the Supplementary Information. Could the authors provide an explanation for why the Sn signals were not observed? Have they attempted longer acquisition times or expanded the chemical shift range to obtain the signals?

Response: We thank the reviewer for this excellent suggestion. These complexes exhibit poor solubility after recrystallization, including in $\text{THF}-d_8$, which resulted in no ^{119}Sn NMR signals being observed for these complexes in solution. We attempted to collect the signals with a 24-hour acquisition time for the ^{119}Sn NMR spectrum of complex **2**, but this was also unsuccessful. Therefore, we examined the solid-state ^{119}Sn NMR spectra for these complexes. Fortunately, the solid-state ^{119}Sn NMR experiments for complexes **1** and **2** yielded signals with large ^{119}Sn chemical shift anisotropy, whereas no signals were detected for complexes **3**, **4**, and **5** despite trying various conditions in the solid state. The solid-state ^{119}Sn NMR spectra for complexes **1** and **2** have been added as Supplementary Figure S4 and Figure S8 in the revised Supplementary Information (SI), respectively. The ^{119}Sn NMR data for these complexes is also discussed in the revised manuscript as follows: "Despite numerous attempts, obtaining liquid-state ^{119}Sn NMR spectra for complexes **1** and **2** was unsuccessful. However, solid-state NMR experiments yielded signals with large ^{119}Sn chemical shift anisotropy. For example, in complex **2**, the single-pulse NMR spectrum reveals three sets of signals at -42, -53, and -296 ppm (Fig. S8c). The combined intensity of the first two peaks is

comparable to that of the resonance at -296 ppm. In the $^1\text{H} \rightarrow ^{119}\text{Sn}$ CP-MAS NMR spectrum (Fig. S8a), the intensity of the peak at -296 ppm is significantly enhanced compared to the resonances at -42 and -53 ppm, indicating that this peak is associated with Sn species that have stronger Sn-H dipolar coupling. Therefore, the peak at -296 ppm can be tentatively assigned to the Sn1 and Sn4 atoms, while the peaks at -42 and -53 ppm can be attributed to the Sn2 and Sn3 sites. This is because Sn1 and Sn4 have 8 hydrogen atoms in their third coordination shell, whereas Sn2 and Sn3 have only one hydrogen atom each. The peak splitting of the latter two peaks (approximately 1600 Hz) is likely due to complex J-coupling between the central Sn species, which is consistent with previous literature.⁵⁴ Additionally, this assignment of the peak at -296 ppm to Sn1 and Sn4 species in complex **2** aligns with the observation of strong signals at similar frequencies (centered at -228 ppm) in the single-pulse ^{119}Sn NMR spectrum of complex **1** (Fig. S4).” in the revised version.

Reviewer #1 (Remarks to the Author): In this resubmitted manuscript, the authors did significant changes, to address the critical comments from both reviewers on the previous version. Therefore, acceptance is recommended.

Response: We sincerely thank the reviewer for the positive evaluation of our study.

Reviewer #2 (Remarks to the Author):

1. The fundamental reactivity studies of isolated Sn(0) and Pb(0) complexes is still elusive in this study. Although they have studied the energy profile of the complex 4 with a few small molecules including CO and BCl₃, but no reactivity study has been conducted with complex 2 either. This is very unusual that none of them reacting to anything, which is difficult to understand. As per the DFT calculation, some external energy source might work in this regard.

Response: We thank the reviewer for this excellent suggestion. In fact, we have attempted to study the reactivity of complex 2 with a series of molecules, but no reactions were observed. Following the reviewer's suggestion, we also conducted DFT calculations of complex 2 with CO, THF, and BCl₃. Similar to the reactions with complex 4, these processes are endothermic. The detailed results have been included in the revised supplementary information as Supplementary Figure 27.

2. ¹¹⁹Sn Solid-state NMR of the isolated complexes have been tried. Only complex 1 and 2 have been studied, but complexes 3, 4, and 5 are not observable. The authors need to write an explanation for the clarity.

Response: We thank the reviewer for this suggestion. The differences between complexes 1 and 2, and complexes 3, 4 and 5 are mainly the direct Sn-Sn bonding between the two Sn sites. Lack of such bonding in the latter may lead to a long longitudinal relaxation time (T_1) (tens of seconds), a characteristic commonly observed in many Sn-containing compounds. [ref. Dupree, R. & Smith, M. E. (eds.) Multinuclear Solid-State NMR of Inorganic Materials. Pergamon Materials Series, Vol. 6, Elsevier Science, Amsterdam, 2002.] We have now added related discussion in the main text.

3. The ¹³C{¹H} NMR spectrum of 2 is also not measured. Authors can analyze solid-state ¹³C NMR, if solution-state measurement is not possible.

Response: We thank the reviewer for this suggestion. As recommended, we have added the solid-state ¹³C NMR spectrum of complex 2 in the revised Supplementary Figure 9.

4. The statement, "Attempts to synthesize the complex with a Pb₄ unit by the reaction of N{CH₂CH₂NLiPⁱPr₂}₃ with PbI₂ and subsequent treatment with PbI₂ and KC₈ were unsuccessfully." seems grammatically incorrect.

Response: Thank you for pointing this out. In the revised version, we have corrected the sentence to: "Attempts to synthesize the complex with a Pb₄ unit by reacting N{CH₂CH₂NLiPⁱPr₂}₃ with PbI₂ and subsequent treatment with PbI₂ and KC₈ were unsuccessful."

5. The reference for the level of theory for DFT calculations in the response sheet is 61, and in the manuscript it's 62.

Response: We apologize for this mistake. The correct reference should be 62, as stated in the manuscript. Thank you for bringing this to our attention.

Reviewer #3 (Remarks to the Author): The authors have addressed most of the issues raised in the previous review and made substantial efforts to test the reactivity of Pb(0) and Sn(0) species, though no positive results were obtained. Their explanation, supported by computational analysis, reasonably attributes this to steric hindrance. With these modifications incorporated, I support the acceptance of the manuscript for publication.

Response: We sincerely thank the reviewer for the positive evaluation of our study.